# Prediction of stunting and its socioeconomic determinants among adolescent girls in Ethiopia using machine learning algorithms

**Alemu Birara Zemariam**[1]*, **Biruk Beletew Abate**[1], **Addis Wondmagegn Alamaw**[2], **Eyob shitie Lake**[3], **Gizachew Yilak**[4], **Mulat Ayele**[3], **Befkad Derese Tilahun**[4], **Habtamu Setegn Ngusie**[5]

1 Department of Pediatrics and Child Health Nursing, School of Nursing, College of Medicine and Health Science, Woldia University, Woldia, Ethiopia, 2 Department of Emergency and Critical Care Nursing, School of Nursing, College of Medicine and Health Science, Woldia University, Woldia, Ethiopia, 3 Department of Midwifery, School of Midwifery, School of Midwifery, College of Medicine and Health Science, Woldia University, Woldia, Ethiopia, 4 Department of Nursing, School of Nursing, College of Medicine and Health Science, Woldia University, Woldia, Ethiopia, 5 Department of Health Informatics, School of Public Health, College of Medicine and Health Science, Woldia University, Woldia, Ethiopia

* alexb7298@gmail.com

**Data Availability Statement:** This dataset used for the current study was publicly available as a supplementary file (sup-1).

## Abstract

### Background

Stunting is a vital indicator of chronic undernutrition that reveals a failure to reach linear growth. Investigating growth and nutrition status during adolescence, in addition to infancy and childhood is very crucial. However, the available studies in Ethiopia have been usually focused in early childhood and they used the traditional stastical methods. Therefore, this study aimed to employ multiple machine learning algorithms to identify the most effective model for the prediction of stunting among adolescent girls in Ethiopia.

### Methods

A total of 3156 weighted samples of adolescent girls aged 15–19 years were used from the 2016 Ethiopian Demographic and Health Survey dataset. The data was pre-processed, and 80% and 20% of the observations were used for training, and testing the model, respectively. Eight machine learning algorithms were included for consideration of model building and comparison. The performance of the predictive model was evaluated using evaluation metrics value through Python software. The synthetic minority oversampling technique was used for data balancing and Boruta algorithm was used to identify best features. Association rule mining using an Apriori algorithm was employed to generate the best rule for the association between the independent feature and the targeted feature using R software.

### Results

The random forest classifier (sensitivity = 81%, accuracy = 77%, precision = 75%, f1-score = 78%, AUC = 85%) outperformed in predicting stunting compared to other ML algorithms considered in this study. Region, poor wealth index, no formal education, unimproved toilet

**Funding:** The author(s) received no specific funding for this work.

**Competing interests:** The authors have declared that no competing interests exist.

**Abbreviations:** ARM, association rule mining; AUC, area under the curve; CSA, Central statically agency; DHS, demographic, and health survey; EDHS, Ethiopian demographic, and health survey; ML, machine learning; ROC, receiver operating characteristic curve; SMOTE, Synthetic Minority Oversampling Technique; WHO, World health organization.

facility, rural residence, not used contraceptive method, religion, age, no media exposure, occupation, and having one or more children were the top attributes to predict stunting. Association rule mining was identified the top seven best rules that most frequently associated with stunting among adolescent girls in Ethiopia.

## Conclusion

The random forest classifier outperformed in predicting and identifying the relevant predictors of stunting. Results have shown that machine learning algorithms can accurately predict stunting, making them potentially valuable as decision-support tools for the relevant stakeholders and giving emphasis for the identified predictors could be an important intervention to halt stunting among adolescent girls.

## Introduction

The World Health Organization (WHO) divides adolescence into two phases: late adolescence, which lasts from 15 to 19 years old, and early adolescence, which lasts from 10 to 14 years old [1]. The number of teenagers worldwide is growing; an estimated 1.2 billion individuals, or 16% of the world's population, are anticipated to be adolescents, and approximately 90% of them reside in low- and middle-income countries [2, 3]. Adolescents make up 20–26% of the population in Ethiopia [4]. Sustaining appropriate physical, cognitive, psycho-social, and emotional development, and linear growth needs adequate nutrition since 15–20% of an adult's height is attained during the adolescence phase [2, 5, 6].

Stunting is a vital indicator of chronic undernutrition that reveals a failure to reach linear growth as a result of continuous food restriction and early-life diseases that prevent a person from growing linearly [7]. Due to higher nutritional needs for growth and development including maturation, traditional marriage and early pregnancy, menarche, and sexual development adolescent females are highly susceptible for undernourishment and stunting [7–9]. In maturity, stunting can lead to decreased productivity, impaired social skills, behavioral issues, and metabolic disorders [10, 11].

More significantly, undernutrition has long-term effects, particularly for females, and stunted teenage girls are more likely to give birth to underweight babies who will also be stunted later in life and they run a high risk of dying during pregnancy and labor if their nutritional demands are not satisfied [8, 12]. Therefore, improving teenage nutrition is one way to end the intergenerational cycle of malnutrition and poor health; if this doesn't happen, the vicious cycle will continue because adolescence is the last opportunity to reduce the effects of malnutrition and end the cycle of malnutrition and poor health [13]. Efforts has been done to reduce stunting among adolescent girls include high-quality nutrition education, sustainable food production, investments for the poor, improved healthcare systems, increased coverage, and strategic plans targeting adolescent girls' nutrition [14], but stunting remains a major public health issue among adolescent girls in developing countries, including Ethiopia [15, 16].

Based on previous studies, stunting among adolescent girls was 48% in Bangladesh, 47% in Nepal [17, 18], and 34.2% in India [17]. In Ethiopia, the prevalence of stunting ranges from 15% to 41.8% [19, 20]. Different studies also reported that age groups, poor wealth index, being rural residents, having family size ≥5, working status, educational status, unimproved toilet facility, and unprotected drinking water source were identified as the predictors of

stunting [13, 15, 16, 20]. Besides, Stunting can cast a long shadow over health, paving the way for chronic issues like heightened vulnerability to infections and enduring developmental delays. Furthermore, its ripple effect on cognitive abilities and educational achievements, which can limit the doors of opportunity for those affected. It also affects the wider socioeconomic repercussions that can drain productivity and inflate healthcare costs for both communities and nations [21–23].

In many societies, girls may have less access to nutritious food compared to boys, especially during adolescence. This disparity can lead to chronic undernutrition, resulting in stunting rather than immediate weight loss. Besides, chronic health issues, such as anemia or infections, can contribute to stunting more significantly than they do to underweight or wasting, as these conditions hinder growth over time [24]. Investigating growth, health, and stunting status during adolescence, in addition to infancy and childhood is very crucial. However, available published studies are limited to adolescents and they are more focused on the early childhood age [15, 20, 25]. While numerous studies have focused on young children, our research did not uncover any predictions of stunting specifically among adolescent girls. We believe that adolescent girls are especially vulnerable to the impacts of stunting [26], which can lead to long-term health and developmental issues. Implementing targeted interventions and evidence-based policies can help address the intergenerational burden of malnutrition. Besides, the available studies in Ethiopia have been used the classical statistical analysis [13, 19, 20]. Traditional analysis methods have limitations like linearity assumptions, limited feature selection, and difficulties with high-dimensional data. They rely on previous assumptions or prior knowledge, which hinders the discovery of hidden information. In contrast, machine learning (ML) excels at capturing nonlinear relationships, automated feature selection, handling high-dimensional data, and adapting to new information. ML is a powerful tool for predicting stunting and tackling complex challenges in low-resource settings [27–29].

ML algorithms offer diverse strategies for predicting, classifying, and uncovering patterns in data. The choice of algorithm depends on specific problem requirements, such as data characteristics, problem at hand, and performance criteria. ML is increasingly used in public health, including disease diagnosis, epidemic surveillance, resource allocation, drug discovery, and remote healthcare. It has the potential to revolutionize healthcare and benefit populations in need [30–32]. ML enables analysis of data to identify at-risk children, handle complex interactions, select key contributing factors, improve accuracy, and facilitate continuous learning. By applying ML, researchers and policymakers can gain insights, identify high-risk groups, and implement effective interventions, ultimately reducing stunting rates and improving child well-being in Ethiopia [4, 30].

This study aims to predict stunting among adolescent girls using eight advanced ML algorithms and association rule mining, filling the gap in previous research that focused on limited ML algorithms and specific health outcomes [30–36]. The findings of this study are relevant to governmental and non-governmental organizations aiming to improve the health and nutrition of often neglected adolescent girls in developing countries. They provide evidence for policymakers to plan integrated interventions and programs, preventing stunting and protecting the health of vulnerable subgroups of adolescents.

## Methods

### Design, data source, setting, and periods

We utilized the 2016 Ethiopian Demographic and Health Survey (EDHS) dataset accessed using the website www.measuredhs.com after requesting and getting approval from the DHS program database. The 2016 EDHS is the fourth survey conducted in Ethiopia to collect data

on household and individual characteristics to provide updated information and/or estimates on key demographic and health indicators of the population [16, 37]. The survey took place from January 18 to June 27, 2016, with a multi-stage stratified sampling technique on 645 enumeration areas. The questionnaires were adapted by EDHS from the DHS Program's standard DHS questionnaires. The survey included a nationally representative sample of women (aged 15–49 years) with a total sample size of 15,683 women [16]. Among this, a sub-sample of 3498 women are adolescents aged 15–19 years and they are eligible for the current study; from which a total weighted 3156 adolescent girls were retained for the final analysis. The data for predicting stunting was obtained from a woman's questionnaire. Out of all the participants, we have analyzed 14 different features.

## Population of the study

All adolescent girls aged 15–19 years in Ethiopia were the source populations for this study, whereas all adolescent girls 15–19 years in the selected enumeration areas (EAs) and whose height is recorded were the study populations.

## Study variables and measurements

The outcome of interest for this study was stunting among adolescent girls which was determined by the WHO Z score of anthropometric indicator of height for age < -2 standard deviation [38] and for the analysis purpose the outcome was binary coded as 1 for stunted and 0 for not stunted.

A set of covariates was considered as the possible risk factors for adolescent stunting and extracted from DHS data set based on the previous studies [11, 15, 19, 20, 39, 40] and the WHO conceptual framework on adolescent stunting: context, causes, and consequences [41]. Based on this the predictors included in the current study include wealth index, age of respondents, educational status, region, residency, family size, number of children, occupation, media exposure, religion, current marital status, source of drinking water, type of toilet facility, and current contraceptive use. Age Group: Current age of the women and re-coded in to two categories with values of "0" for 15–17, "1" for 18–19. Religion: Recoded in four categories with a value of "0" for Muslim, "1" for Orthodox, "2" for protestant, and "3" for other religious groups (combining catholic and traditional). Occupation: Re-coded in two categories with a value of "0" for not working, and "1" for working. Media exposure: A composite variable obtained by combining whether a respondent reads newspaper/ magazine, listen to radio, and watch television with a value of "0" if women were not exposed to at least one of the three media, and "1" if a woman has access/exposure to at least one of the three media [42]. Educational status: this is the minimum educational level a woman achieved and coded into four groups with a value of "0" for no education, "1" for primary education, and "2" for secondary education, and "3" for higher education. Family size: Recoded in to three categories as 1–3, 4–6, and seven and above. The five quintiles wealth index were re-categorized as poor, medium, and rich. Items about drinking water and sanitation were based on the core questions on the source of drinking water and sanitation for household surveys developed by the WHO and DHS guide and it was re-coded into two categories as "unprotected" and "protected source of drinking water" and "improved" and "unimproved" toilet facility [16, 43, 44].

## Data preprocessing and analytic strategies

Data pre-processing is a vital task before developing a prediction model and has a significant impact on the model prediction performance. Data preparation techniques includes data

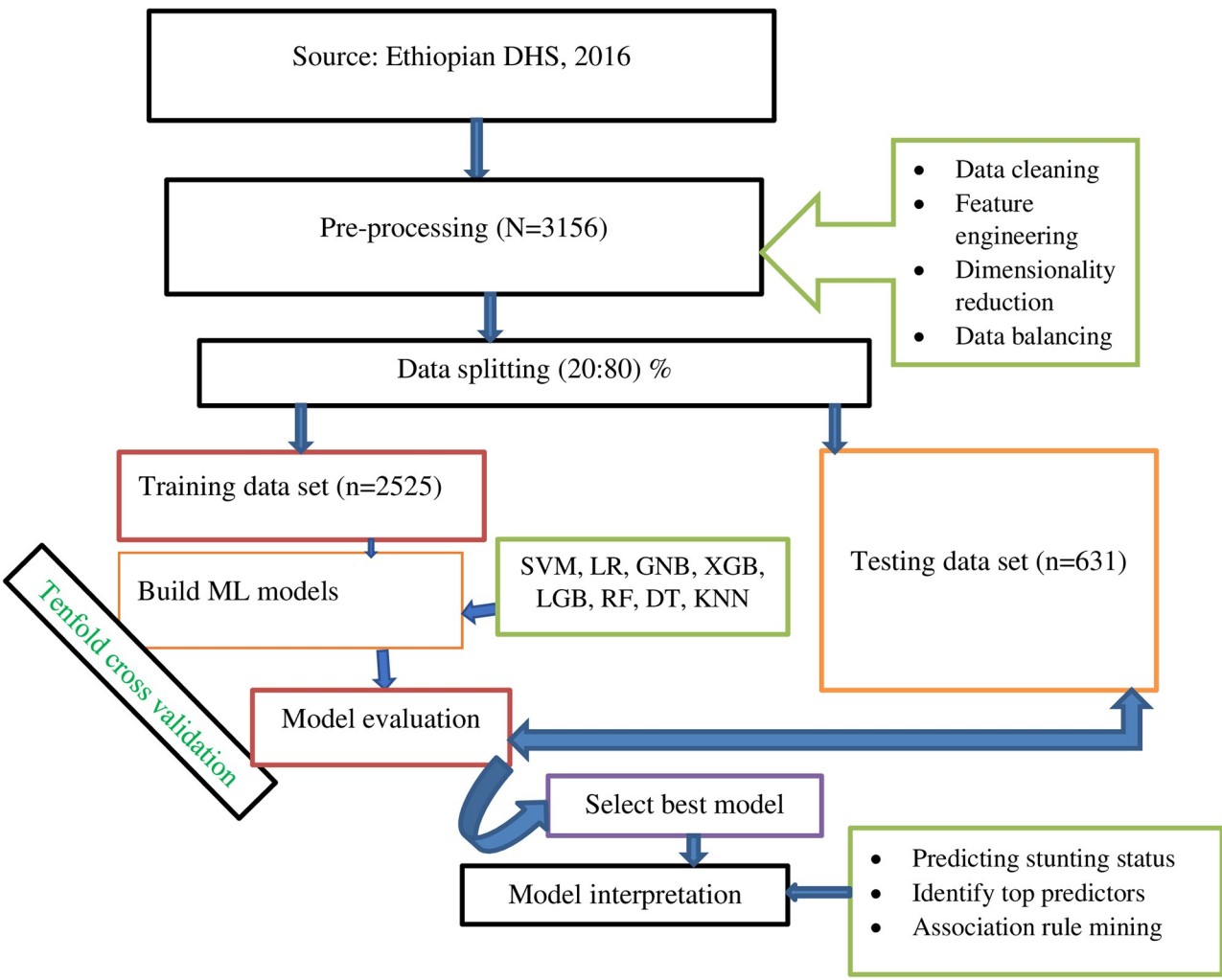

**Fig 1. Study workflow diagram.**

cleaning, feature engineering, dimensionality reduction, and data splitting [45]. The detail of the workflow for the current study is presented (see Fig 1).

## Data cleaning

Data cleaning is the first crucial step performed after the data were retrieved consisting of detecting and removing outliers, handling missing values, and handling unbalanced categories of the outcome variable from the data. We explored different methods of missing data management in ML, such as deletion, imputation, model-based imputation, and domain-specific knowledge. Considering factors like missingness nature, missed data amount, assumptions, and the ML algorithm used, we have opted to handle missing values in our dataset using K-nearest neighbor (KNN) imputation. KNN imputation retains all data, handles outliers, works for numerical and categorical features, adapts to new data, and reduces bias while encompassing a wide range of values. We identified outliers through scatter plots, box plots, and histograms, and assessed multicollinearity by examining the correlation matrix. A correlation value above 0.8 indicated high correlation between variables [46].

## Imbalanced data handling

ML models trained on imbalanced data are typically biased toward the majority class and fail to predict cases that are rare/minority class [47]. To address this issue, researchers have developed various mechanisms. This study employed four balancing methods: under-sampling, over-sampling, ADASYN, and SMOTE [48]. After training our ML algorithms on unbalanced data, we experimented with these techniques to select best data balancing technique. We evaluated model performance using accuracy and AUC metrics and found that the SMOTE technique [49] outperformed the others, making it the chosen approach for data balancing in the final model prediction.

## Feature engineering

Feature engineering is the process of transforming raw data into features that better represent the underlying problem to predictive models. Among various feature engineering techniques, the classical one-hot coding method was utilized for encoding categorical variables into numeric values, and label encoding for coding each category of variables as a number was done through the preprocessing module of the scikit-learn package.

Additionally, dimensionality reduction was conducted to decrease the number of input variables for the predictive model, aiming to create a simpler and more effective model for making predictions on new data. There are two approaches to dimension reduction: feature selection and feature extraction, with the latter being more appropriate for image processing [50]. Feature selection involves choosing the most relevant independent variables that have the greatest impact on predicting the target variable. Feature selection is the appropriate method for our dataset, while feature extraction is typically utilized for datasets involving image processing and deep learning. We have examined different feature selection techniques, including Lasso, Principal component analysis, forward selection, backward elimination, recursive feature elimination, correlation-based feature selection, and chi-square test. Their performance was evaluated using evaluation metrics [51]. Therefore, the result revealed that Boruta is the most effective method for feature selection in our dataset. Boruta is a wrapper-based technique and known for its unbiased and consistent performance, making it highly effective in selecting key variables for model prediction [52, 53]. Combining Boruta with the random forest classifier offers benefits such as improved feature selection, noise and irrelevant feature robustness, reduced bias in feature importance, reduced over fitting, resulting in better model performance, and enhanced interpretability. However, challenges and limitations associated with these techniques were existed. To address the limitations, we employ techniques such as L1 or L2 regularization, cross-validation, independent test sets, parallel processing, analyzing feature importance stability, recursive feature elimination, balancing false positives and false negatives, and conducting principal component analysis [54].

Data splitting:- to train the model and validate it on data it has never seen before a simple 80/20 split method in which 80% of samples (2525 respondents data) were used for testing and the rest 20% of respondents (631 sample) used for testing the model. Besides, a tenfold cross-validation method was used in this study for model training as it does not waste a lot of data, which is a big advantage when the number of samples is small [55].

## Model selection

After the data had been divided into training and testing tests, suitable models were selected to perform the training. Since the outcome variable was categorical, the task was a classification task and appropriate classifiers needed to be selected to conduct the prediction. The dataset used in the analysis falls under the category of binary classification since stunting was

categorized into two mutually exclusive categories. We have applied eight ML algorithms used for this analysis logistic regression (LR), Random forest (RF), K-neigh rest neighbor (KNN), support vector machine (SVM), Gaussian Naïve Bayes (GNB), eXtreme gradient boosting (XGBoost), decision tree (DT) and light gradient boost (LGB) classifiers. These algorithms were selected based on previous studies that applied ML techniques for classification tasks on EDHS data [56–59]. Furthermore, the choice of these algorithms was determined based on their scalability, interpretability, number of features, computational efficiency, characteristics of the data, problem type, robustness to noise and outliers, accuracy, bias-variance trade-off, and domain expertise.

In this study, we utilized the scikit-learn version 1.3.2 packages in Python, implemented within Jupyter Notebook, to employ ML algorithms. The descriptions of each algorithm are as follows:

A) Decision tree (DT)

A DT is a non-parametric technique that classifies a data set based on the problem's predictive structure. It produces a classification tree for categorical variables. Decision trees are highly interpretable, efficiently capture nonlinear relationships, handle both categorical and numerical features, relatively robust to outliers and noisy data, handle missing values by utilizing surrogate splits or imputation techniques, and can handle large datasets efficiently [60]. However, DT also has limitations as they can be prone to over fitting, struggle with capturing certain complex relationships that require more sophisticated algorithms, and can be sensitive to small changes in the data, leading to different tree structures.

B) Random forest (RF)

RF is a type of supervised ML that can be used for classification, regression, and dimension reduction purposes. It is a versatile algorithm used for huge amounts of data and overcoming noise. RF is preferred when improved predictive performance, reduced bias, reduction of variance, robustness to noise and outliers, feature importance, and handling high-dimensional data are important considerations for the problem at hand [34, 61]. However, RF has some limitations. It can be a black-box model, making it less interpretable or more difficult to interpret compared to individual DT; the ensemble nature of random forests makes it challenging to trace the decision-making process. Additionally, RF may not perform well on datasets with strong linear relationships.

C) Extreme gradient boost (XG Boost)

XG boost is a DT-based ensemble machine learning algorithm working by a gradient boosting framework. Boosting involves combining weak classifiers to produce a powerful averaged classifier, and it is also a variance reduction technique. It can be applied to both classification and prediction problems. XG boost is preferred because of robust to noisy data and outliers, handle high-dimensional datasets, control model complexity and prevent over fitting, handle missing values in the data, saves computational resources, and provides a wide range of hyper parameters [62]. However, XG boost may have higher computational and memory requirements and it also tends to be less interpretable compared to the other algorithms.

D) Light gradient boosting machine (LGM boost)

Light GBM is a gradient-boosting framework that works by combining multiple learners usually DT to create a strong predictive model and reduce memory usage. Light GBM is generally faster and more memory-efficient, making it suitable for large datasets than XG boost [63]. Light GBM is preferred when efficiency, scalability, handling high-dimensional data, handling

categorical features, advanced boosting techniques, regularization techniques, feature importance, handling imbalanced datasets, and flexibility are important considerations for the problem at hand.

E) Support vector machine (SVM)

SVM is a set of supervised learning methods used for classification, regression, and outlier detection. SVMs is preferred when dealing with high-dimensional spaces, robustness to outliers, nonlinearity, margin maximization, memory efficiency, and small to medium-sized datasets are important considerations for the problem at hand [64]. However, SVMs may have limitations in terms of scalability to large datasets and computational efficiency, especially when using non-linear kernels. Besides, SVMs may not perform well when the dataset is imbalanced, or when the classes are overlapping and not well-separated. Moreover, SVM do not provide probability estimates directly but through an expensive five-fold cross-validation process

F) Logistic regression (LR)

LR is a supervised ML algorithm used to solve classification issues. It is a parametric method that assumes a Bernoulli distribution of the target variable and the independence of the observations [64].

G) K-nearest Neighbor (KNN)

KNN is a non-parametric, robust, and adaptable supervised ML primarily used for classification problem. This approach keeps track of all existing cases and categorizes new ones using a similarity score with a distance function and the majority vote of its neighbors. KNN is preferred when dealing with nonlinear relationships, interpretability, robustness to outliers, handling imbalanced datasets, no explicit training step, flexibility, and datasets with varying densities are important considerations for the problem at hand [65]. However, KNN has limitations. It can be computationally expensive, especially when dealing with large datasets or high-dimensional feature spaces. Besides, KNN is sensitive to the choice of the distance metric, and the optimal value of K needs to be determined through experimentation or cross-validation.

H) Gaussian Naïve Bayes (GNB)

NB is a collection of ML algorithms built based on Bayes theorem which has two basic assumptions. The first one is every pair of features should be independent of each other and the second assumption is the feature must have an equal contribution to the outcome prediction. GNB is preferred when efficiency, simplicity, handling continuous features, small training sets, text classification, and the feature independence assumption are important considerations for the problem at hand [66]. However, GNB may not perform well in cases where the two assumptions are severely violated. It may struggle with datasets where the features have strong dependencies or when the decision boundary is complex.

## Model training and evaluation

After dividing the data into training and testing sets, we selected appropriate models for training, focusing on classifiers suitable for the categorical target variable. The dataset involved binary classification for anemia, so we utilized eight machine learning algorithms. These choices were based on previous research using machine learning techniques on EDHS data, the type of problem, and data nature or characteristics.

Following model selection, we trained the selected classifiers with both balanced and unbalanced data. The best predictive model was then chosen and trained with balanced training data for the final prediction on unseen test data. To evaluate the performance of the final model, we used a confusion matrix and receiver operating characteristic (ROC) curve with metrics such as accuracy, recall, specificity, F1 score, and area under the curve (AUC). The AUC was considered the main performance metric, providing an overall assessment of the model's performance at different classification thresholds. The confusion matrix allowed us to extract one-dimensional performance metrics such as True Positive (TP), True Negative (TN), False Positive (FP), and False Negative (FN) [47].

When selecting an evaluation metric, we have considered contextual requirements, metric trade-offs, field benchmarks, model interpretability, problem type, data characteristics, and task goals [67–69]. In addition to standard metrics, the model's performance was further evaluated using tenfold cross-validation. This technique involves dividing the data into ten subsets and training and evaluating the model ten times with different combinations of subsets [70]. The study also conducted a comprehensive exploration of hyper parameters to optimize the model's performance. Grid search, random search, and Bayesian optimization methods were systematically utilized to find the most effective hyper parameter configurations, considering factors such as search space size, computational resources, and exploration-exploitation balance. Grid search is exhaustive but computationally expensive, random search is less intensive but may require more iteration, and Bayesian optimization is efficient for complex search spaces but requires additional setup and resources. The choice of method depends on the specific algorithm and dataset characteristics, and experimentation on the validation set is recommended for tuning [71]. The authors considered the advantages of each technique and selected the best tuning approach based on performance metrics. Moreover, calibration was performed to enhance the precision and reliability of the model, resulting in improved prediction accuracy.

## Association rule mining

In this study, association rule analysis was applied through the Apriori algorithm (arules package) via R software (version 4.3.2) to identify a specific category of predictor variables that have associations with stunting. The association rule mining analysis (the If (antecedent)/ then (consequent) statements) was used to discover relationships between seemingly relational attributes, especially for the categorical nature of attributes because the ML algorithms do not show which category is more associated with stunting among adolescent girls in Ethiopia. It is important to observe frequently occurring patterns and identify the dependencies between attributes by supporting how frequently the if/then relationship appears in the observations and confidence in the number of times the relationships are true [72]. The If then association rule is the pair of X and Y (X, Y) attributes expressed as X->Y, where X is an antecedent and Y a consequent that is as X happens Y would also happen. The relationship between X and Y attributes is expressed in the following way [73]. As lift is equal to 1, it shows that X and Y appearing at the same time belong to independent random events and have no special significance; we call these rules an uncorrelated rule. If the lift value is less than 1, it shows that the occurrence of "X" reduces the occurrence of "Y," and then we call them negative correlation rules. If the lift value is greater than 1, it shows that the occurrence of "X" promotes the occurrence of "Y," and then we call them positive correlation rules.

## Model interpretability

Researchers have highlighted the inclusion of SHAP (SHapley Additive exPlanations) values to comprehend how different features influence the model's predictions, SHAP analysis emerges

as a more appropriate choice [74, 75]. In order to comprehensively understand the data and analyze the factors impacting stunting prediction, we employed various techniques. Firstly, we utilized average SHAP values to evaluate the overall impact of each feature on the model's predictions. This approach provided insights into the relative importance of different variables. SHAP analysis is a commonly used method in machine learning for interpreting predictions and understanding feature importance. It assigns a numerical value, known as a SHAP value, to each feature, indicating its contribution to predictions. Calculating SHAP values allows practitioners to gain insights into how features influence predictions. Positive values indicate a positive contribution, negative values indicate the opposite, and the magnitude represents the strength of influence. SHAP analysis enhances transparency and interpretability, providing a global view of feature importance and explaining individual predictions [76–78]. Afterward, we employed a waterfall plot to visually depict the cumulative effects of these variables, emphasizing their contributions to the overall prediction [79].

## Ethical considerations

The central statistical agency (CSA) received the ethical clearance for the 2016 EDHS survey from the Ethiopian Health and Nutrition Research Institute Review Board and the National Research Ethics Review Committee at the Ministry of Science and Technology. Moreover, they confirmed that their research has been performed in accordance with the declaration of Helsinki and the CSA obtained written informed consent from the respondents. The authors received a permission letter to download and use the data set for the current study from the DHS program data archivist upon submission of a proposal. After data access was authorized by DHS we maintained confidentiality and used only for the study purpose.

## Results

### Study participant characteristics

A total of weighted 3156 adolescent girls were included in the final analysis. Of these, the prevalence of stunting was found to be 452 (14.32%) with 95% CI; 13.12, 15.59. The majority (60.17%) of the respondents were in the age group of 15–17 years. Nearly two-thirds (64.5%) of the adolescent girls were rural dwellers and 58.24% had completed primary education. Four hundred two (12.74%) and one-hundred fifty-three (4.85%) adolescent girls were from Addis Ababa and Harari regions, respectively. The majority (69.8%) of the adolescent girls did not have access to improved toilet facilities and 44.3% had no media exposure. The majority (89.6%) of the respondents had no child and four out of ten (41.13%) respondents had four to six family sizes (see Table 1).

### Machine learning analysis of stunting among adolescent girls

**Data balancing.** We had employed four data balancing techniques such as under-sampling, over-sampling, SMOTE, and ADASYN, and their performance was assessed using an accuracy and AUC value. The balancing techniques that demonstrated high performance were considered as best balancing technique for the final prediction. In terms of unbalanced data, the LGB achieved an AUC value of 64% and among the four balancing technique, the RF algorithm outperformed than the other algorithms with an AUC value of 84%. Considering all the data balancing techniques, SMOTE stood out as the superior method. Table 2 depicted a comparison of different data balancing techniques with an AUC and accuracy value. Moreover, the prevalence of stunting before and after data balancing was reported (Fig 2).

**Table 1. Socio-demographic characteristics among adolescent girls in Ethiopia, 2016 EDHS.**

| | | Stunting | | Weighted sample = 3156 |
|---|---|---|---|---|
| Variables | Categories | Yes (n) | No (n) | Frequency n (%) |
| Age in years | 15–17 | 223 | 1676 | 1899 (60.17) |
| | 18–19 | 229 | 1028 | 1257 (39.83) |
| Residence | Urban | 158 | 962 | 1120 (35.49) |
| | Rural | 294 | 1742 | 2036 (64.51) |
| Educational status | No formal education | 93 | 425 | 518 (16.41) |
| | Primary | 256 | 1582 | 1838 (58.24) |
| | Secondary | 88 | 623 | 711 (22.53) |
| | Higher | 15 | 74 | 89 (2.82) |
| Region | Tigray | 68 | 323 | 391 (12.39) |
| | Afar | 54 | 190 | 244 (7.73) |
| | Amhara | 61 | 260 | 321 (10.17) |
| | Oromia | 62 | 325 | 387 (12.26) |
| | Somali | 12 | 275 | 287 (9.09) |
| | Benishanguel Gumez | 23 | 184 | 207 (6.56) |
| | SNNPR | 44 | 315 | 359 (11.38) |
| | Gambella | 7 | 179 | 186 (5.89) |
| | Harari | 15 | 138 | 153 (4.85) |
| | Addis Ababa | 84 | 318 | 402 (12.74) |
| | Dire Dawa | 22 | 197 | 219 (6.94) |
| Marital status | Never in union | 334 | 2107 | 2441 (77.34) |
| | Married | 102 | 483 | 585 (18.54) |
| | Living with partner | 3 | 17 | 20 (0.63) |
| | Divorced/separated | 13 | 97 | 110 (3.49) |
| Religion | Orthodox | 216 | 1064 | 1280 (40.56) |
| | Muslim | 173 | 1109 | 1282 (40.62) |
| | Protestant | 60 | 495 | 555 (17.59) |
| | Others* | 3 | 36 | 39 (1.24) |
| Occupation | Not working | 262 | 1719 | 1981 (62.77) |
| | Working | 190 | 985 | 1175 (37.23) |
| Source of drinking water | Unprotected | 139 | 787 | 926 (29.34) |
| | Protected | 313 | 1917 | 2230 (70.66) |
| Type of toilet facility | Unimproved | 303 | 1899 | 2202 (69.77) |
| | Improved | 149 | 805 | 954 (30.23) |
| Current contraceptive use | No | 423 | 2541 | 2964 (93.92) |
| | Yes | 29 | 163 | 192 (6.08) |
| Number of children | No child | 394 | 2435 | 2829 (89.64) |
| | 1 and above | 58 | 269 | 327 (10.36) |
| Family size | Three and below | 105 | 599 | 704 (22.31) |
| | Four to six | 175 | 1123 | 1298 (41.13) |
| | Seven and above | 172 | 982 | 1154 (36.57) |
| Media exposure | Yes | 258 | 1501 | 1759 (55.74) |
| | No | 194 | 1203 | 1397 (44.26) |
| Wealth index | Poor | 159 | 935 | 1094 (34.66) |
| | Medium | 70 | 352 | 422 (13.37) |
| | Rich | 223 | 1417 | 1640 (51.96) |

Note: others*-traditional, catholic, EDHS-Ethiopian demographic and health survey, SNNPR-south nation national people representative

**Table 2. Comparison of imbalanced data handling techniques using accuracy and AUC values.**

| Algorithms | Evaluation metrics | Unbalanced data | Under sampling | Over sampling | SMOTE | ADASYN |
|---|---|---|---|---|---|---|
| SVM | Accuracy | 86 | 54 | 66 | 64 | 64 |
| | AUC | 52 | 57 | 70 | 69 | 69 |
| GNB | Accuracy | 85 | 62 | 59 | 57 | 58 |
| | AUC | 59 | **67** | 63 | 60 | 63 |
| LR | Accuracy | 86 | 61 | 61 | 60 | 61 |
| | AUC | 61 | 66 | 65 | 63 | 65 |
| DT | Accuracy | 80 | 47 | 73 | 73 | 70 |
| | AUC | 53 | 46 | 76 | 79 | 74 |
| RF | Accuracy | 83 | 54 | 75 | 76 | 72 |
| | AUC | 57 | 54 | **79** | **84** | **78** |
| LGB | Accuracy | 85 | 52 | 67 | 64 | 65 |
| | AUC | **64** | 52 | 72 | 69 | 71 |
| XGB | Accuracy | 82 | 49 | 73 | 74 | 70 |
| | AUC | 59 | 50 | **79** | 80 | 76 |
| KNN | Accuracy | 85 | 53 | 65 | 62 | 63 |
| | AUC | 56 | 52 | 70 | 68 | 69 |

Note: LGBM-light gradient boosting machine, GNB-Gaussian Naive Bayes, AUC-area under curve, KNN- k-nearest neighbor, SMOTE, synthetic minority oversampling technique, SVM, support vector machine, DT-decision tree, XGB-extreme gradient boosting, RF-random forest, LR-logistic regression, ML-machine learning.

**Features selection using Boruta algorithms.** The important features from the data set were selected by using the Boruta algorithm, which classifies the independent variables as either important or unimportant based on their impact on the stunting status. The algorithm identified the most influential confirmed key features that can explain the variation in stunting status and recommended them for further analysis and modeling. On the other hand, variables that were rejected are considered less important and were excluded from further analysis, as they have been determined to have minimal impact on the outcome of interest. The Boruta algorithm graph showed the confirmed (important) variables with green color. The rejected (unimportant) variables were represented with red color [52]. From a total of 14 features, the three features namely family size, marital status, and source of drinking water were considered unimportant and the rest 11 features are important for model prediction (Fig 3).

**Model development and performance evaluation to predict stunting.** Performance metrics such as accuracy, precision, recall, F1 score, specificity, and AUC value were used to evaluate and compare the algorithms' performance (Fig 4). These metrics assessed the overall correctness, ability to correctly predict positive and negative instances, and the algorithm's discriminative power. By utilizing these performance metrics, the researchers conducted a comprehensive evaluation to determine how effectively the algorithms could predict stunting among adolescent girls in Ethiopia. After comparing the performance metrics of the three tuning techniques we found that the grid search was the best tuning technique (Table 3). Based on the ROC curve analysis result, the top three ML algorithms for classifying stunting status were found to be the random forest, the light gradient, and the extreme gradient boosting classifier (Fig 5).

A

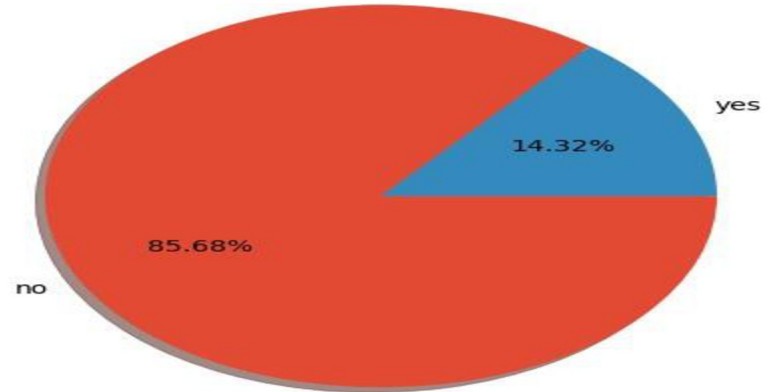

B

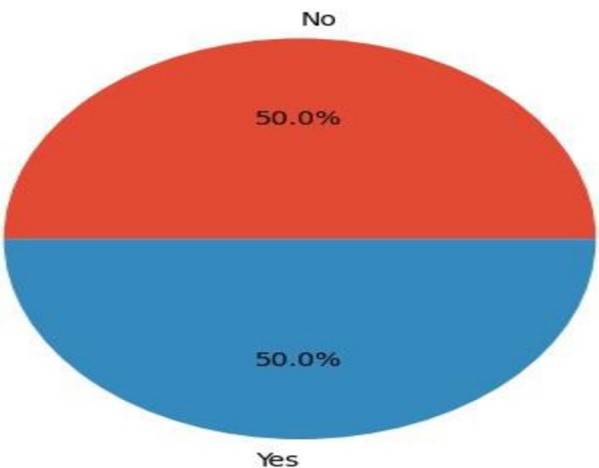

**Fig 2. Prevalence of stunting among adolescent girls in Ethiopia, 2016 with unbalanced data (A) and balanced date using SMOTE (B).**

### Model interpretability

**SHAP value interpretation.** The mean SHAP value report offered valuable insights into the comparative significance of various features in the classification model. Region, respondent age, educational status, and wealth index were identified as the most influential factors, exerting a substantial impact on the model's predictions. Besides, assume the baseline log odds for stunting without considering the region is 0. A SHAP value of +0.04 suggests that the region and respondent age increases the log odds of stunting. This means that the presence of this

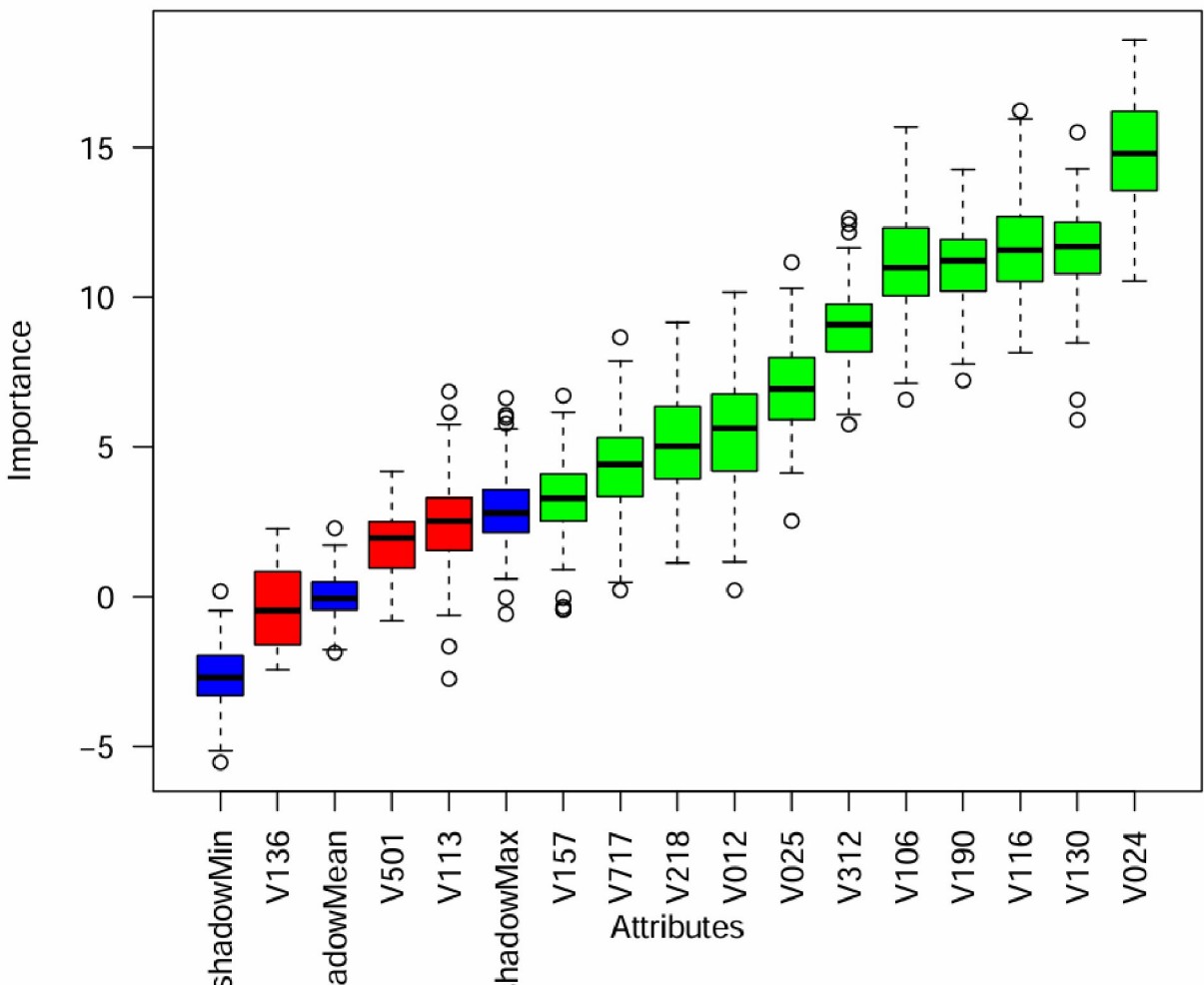

**Fig 3. Feature selection using the Boruta algorithm method.** Note: V024-region, V130-religion, V190-wealth index, V116-toilet facility type, V312-contraceptive method, V025-residence, V157-media exposure, V106-educational status, V218-number of children, V501-marrital status, V717-occupation, V113-source of drinking water, V136-family size, and V012-age of respondent.

region and respondent age increases the probability of stunting to around 51%. Conversely, the contraceptive use and residence displayed minimal influence and limited value in predicting the model's outcome on the classification outcome, as evidenced by their low mean SHAP values. A SHAP value of +0.01 for residence and contraceptive use indicates a very slight increase in the likelihood of stunting, suggesting that while these factors have some relevance, their impact is minimal compared to other potential factors influencing child growth. Moreover, A SHAP value of +0.01 implies that both residence and contraceptive use have a small positive effect on the likelihood of stunting. This indicates that the likelihood of stunting increases slightly for about 50.25% (Fig 6).

The waterfall plot provided valuable insights into the hierarchy of feature importance when predicting the target variable. The plot highlighted that region, educational status, and respondent age had the highest positive impact on the prediction. A SHAP value of +0.07 is relatively substantial, indicating that the region plays a noteworthy role in increasing the likelihood of

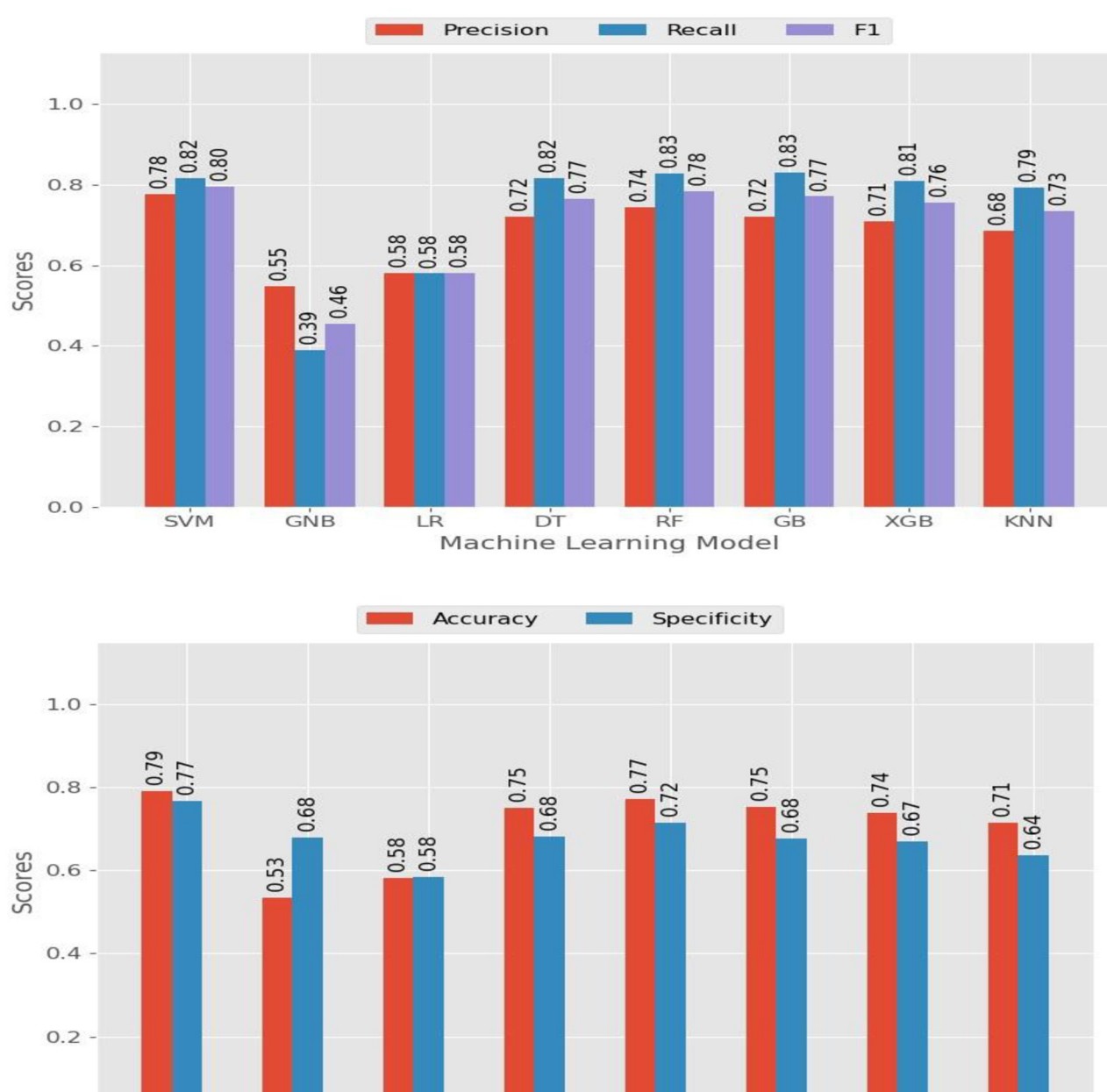

**Fig 4. Performance evaluation measure of the machine learning algorithm after data balancing with SMOTE and grid search tuning.**

stunting. This suggests that the probability of stunting increases to about 54.2% due to the regional factor. Contrariwise, number of children, media exposure and respondent occupation had had a negative contribution to the model's prediction meaning that they had a decreasing effect on the prediction. A SHAP value of -0.02 suggests that having more children has a negative effect on the likelihood of stunting. This indicates that the probability of stunting decreases slightly for about 52% in families having fewer children compared to those who have more number of children (Fig 7).

**Table 3. Accuracy and AUC value of ML algorithms using three hyper parameter tuning techniques.**

| Algorithm | Grid search | | Random search | | Bayesian optimization | |
|---|---|---|---|---|---|---|
| | Accuracy | AUC | Accuracy | AUC | Accuracy | AUC |
| SVM | 79 | 0.89 | 69 | 0.79 | 69 | 0.65 |
| GNB | 53 | 0.68 | 56 | 0.63 | 70 | 0.72 |
| LR | 58 | 0.71 | 59 | 0.66 | 70 | 0.72 |
| DT | 75 | **0.91** | 71 | 0.84 | 65 | 0.68 |
| RF | 77 | **0.95** | 75 | 0.90 | 68 | 0.70 |
| LGB | 75 | **0.93** | 73 | 0.87 | 64 | 0.65 |
| XGB | 74 | **0.91** | 74 | 0.87 | 68 | 0.70 |
| KNN | 71 | 0.90 | 70 | 0.86 | 61 | 0.64 |

**Association rule mining.** The association rule mining was applied and a total of 29 association rules were built. Among these, the authors selected the top 7 best rules based on their confidence and lift value [80], which is an interesting quality measurement criterion of the association rule. To identify all potential association rules, the minimal support degree was set at 0.0001 and the minimum confidence threshold at 80% because a rule is considered reliable if its confidence level is more than 80% [81]. These factors significantly influence the probability of stunting and should be considered in initiatives aimed at enhancing adolescent health in the region. The top seven association rules are presented below:

Rule-1: If adolescent girls age = 1(aged 18-19yrs), religion = 1(orthodox), wealth index = 0 (poor), number of children = 1 (at least 1 and above child), and current contraceptive use = 0 (not used), Then the probability of being stunted is 95.8% (lift = 6.9).

Rule-2: If adolescent girls age = 1 (aged 18-19yrs), education level = 0 (no formal education), religion = 1 (orthodox), wealth index = 0(poor), current contraceptive use = 0 (not used), Then the likelihood of being stunted is 95.8% (lift 6.9).

Rule-3: If region = 3 (Amhara), wealth index = 0 (poor), numbers of children = 1(at least 1 and above child), contraceptive use = 0 (not used), media exposure = 0(no), Then the likelihood of stunting is 83.3% (lift = 5.8).

Rule-4: If region = 10 (Addis Ababa), type of toilet facility = 0(unimproved), number of children = 1(at least 1 and above child), Then the likelihood of stunting is 80% (Lift = 5.6).

Rule-5: If region = 2 (Afar), type of toilet facility = 0(unimproved), respondent occupation = 1 (working), media exposure = 0(no), Then the likelihood of stunting is 80% (Lift = 5.6)

Rule-6: If region = 1(Tigray), residence = 2(rural), type of toilet facility = 0 (unimproved), number of children = 1(at least 1 and above child). Then the probability of stunting would be 80% (lift = 5.6)

Rule-7: If respondent age = 1 (aged 18-19yrs), region = 1(Tigray), type of toilet facility = 0 (unimproved), number of children = 1(at least 1 and above child), Then the likelihood of being stunted is 80%, (lift = 5.6)

## Discussion

The findings of this research demonstrated the potential of ML algorithms in predicting the presence of stunting among adolescent girls in Ethiopia. This opens up opportunities for the

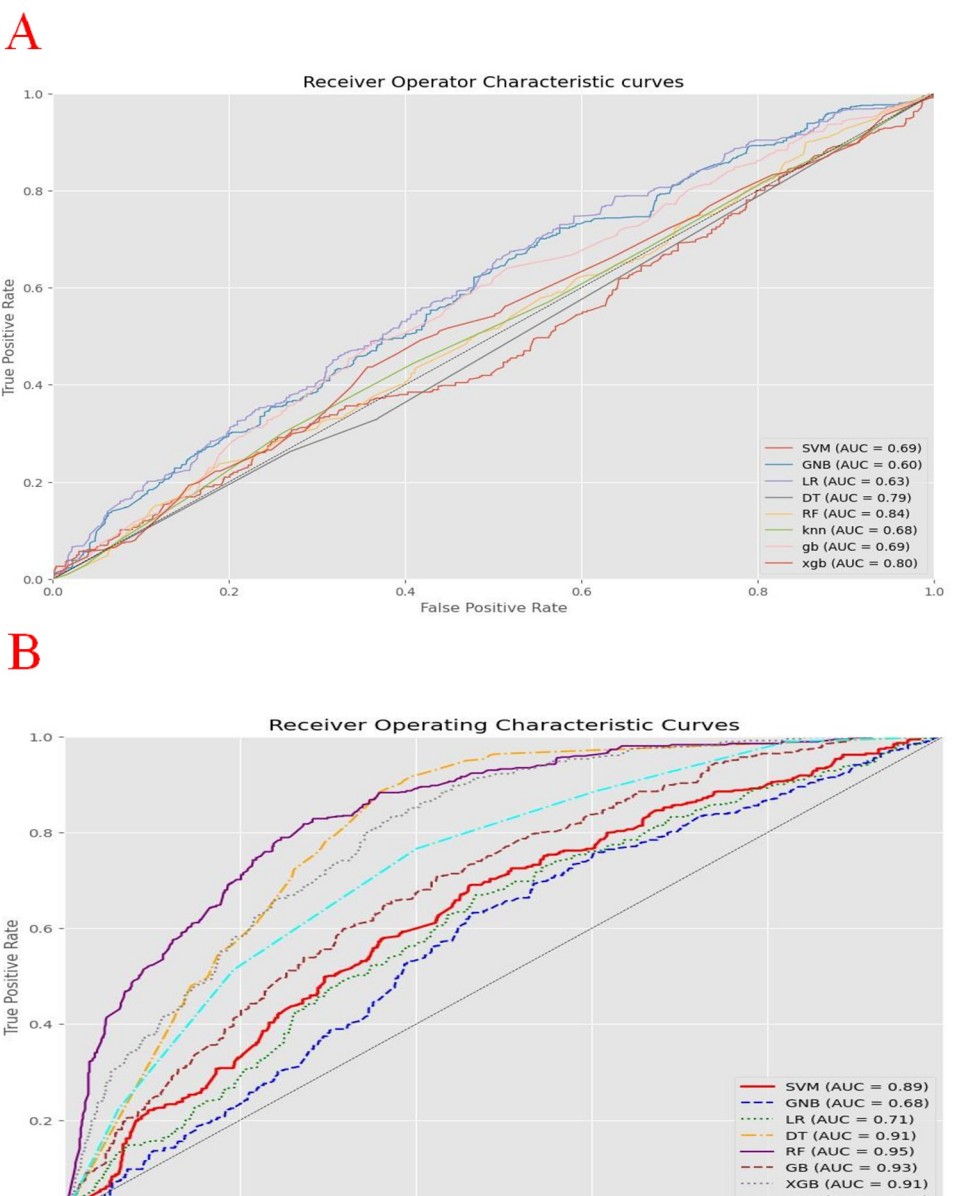

**Fig 5. ROC curve analysis of machine learning algorithms with balanced data before tuning (A) and after optimized with hyper parameter tuning (B).**

development of automated screening tools and decision support systems that can assist health-care providers in diagnosing and managing stunting. We have utilized eight different ML algorithms, namely RF, DT, GNB, KNN, LGB, XGB, SVM, and LR to assess and compare their predictive capabilities. Evaluating their performance we found that all eight algorithms achieved ROC values above the optimal threshold and the RF algorithm performed better than all others, with an accuracy of 77% and an AUC value of 95%. Although there were slight differences in the metrics values due to socio-economic, size of data, and study area variations, the finding of the current study was similar with studies conducted in Bangladesh [82], Zambia

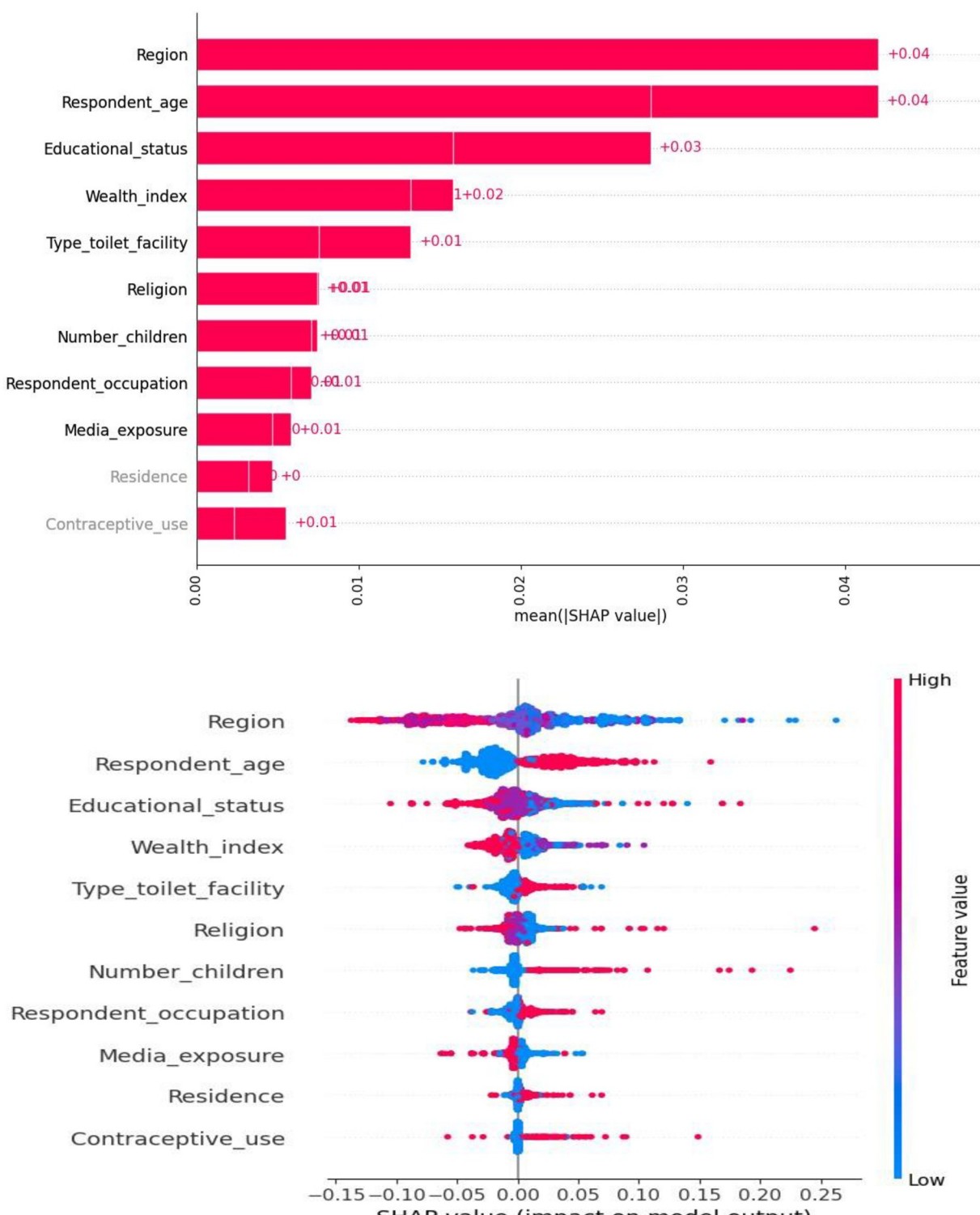

**Fig 6. A mean SHAP value report.**

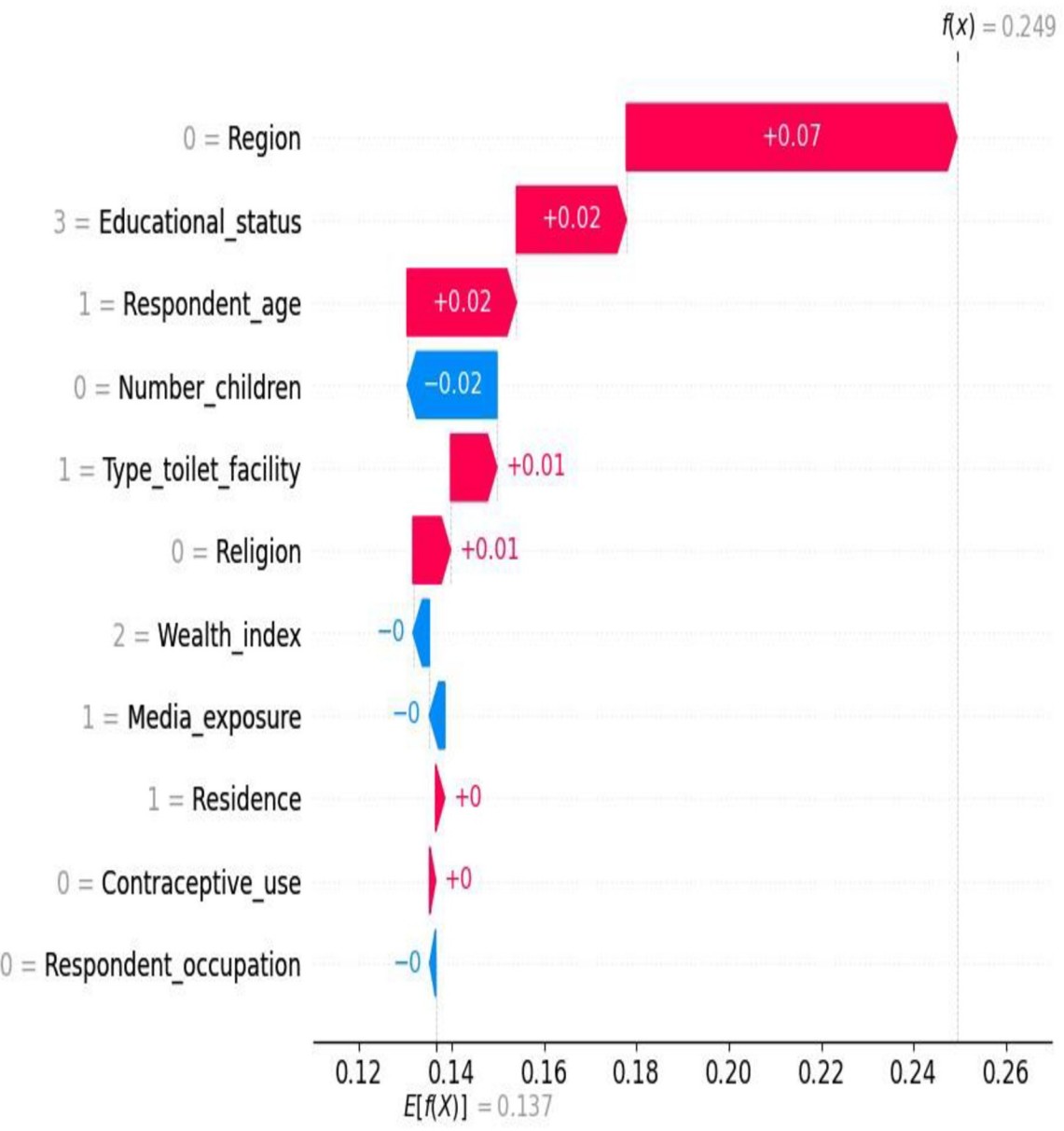

**Fig 7. A waterfall plot analysis.**

[64], and Ethiopia [57], which also found that the RF model was the best than the other algorithm. This similarity might be due to the nature of the features included across different studies since the RF algorithm demonstrated better performance in categorical variables, high dimensions and non-linear trends, and requires minimal effort for tuning hyper-parameters [82]. The use of RF classifier for predicting stunting has implications by providing accurate predictive models, insights into risk factors and mechanisms, identification of vulnerable subgroups, and the potential for integrating machine learning into healthcare systems. These implications pave the way for targeted interventions, personalized healthcare approaches, and improved health outcomes for individuals affected by stunting. However, our study was

incongruent with a studies conducted in Papua NewGuinea [25], Ethiopia [35], and Rwanda [83], which showed that the XGB has a superior performance compared to all other algorithms. This discrepancy might be due to the nature of data, population characteristics, different evaluation metrics criteria used across studies, and research bias arising in the sample size, data preprocessing techniques, feature engineering that affect the observed results and model prediction [30].

Another aim of this study was to identify the top predictors of stunting among adolescent girls. To accomplish this, the author utilized the Boruta algorithm to select important features. Out of a total of 14 features included based on the literature; the study identified the top 11 predictors of stunting. The Boruta algorithm revealed that region, religion, type of toilet facility, wealth index, educational level, and current contraceptive use, and residence, age of respondent, number of children, occupational status, and media exposure were found to be the top 11 predictors of stunting among adolescent girls in Ethiopia.

The finding obtained from analyzing the mean SHAP value report and waterfall plot provided valuable insights regarding the importance of different factors in a classification model used to predict stunting. Factors such as region, respondent age, wealth index and educational status were found to have a significant impact on the model's predictions and emerged as the most influential features. Understanding that certain regions have a higher SHAP value for stunting can guide policymakers and health professionals to focus interventions in these areas and programs aimed at improving nutrition, health education, and resources could be prioritized in regions identified as having higher risks.

On the other hand, the contraceptive use and residence had minimal influence on the classification outcome, as indicated by their low mean SHAP values. Understanding the significance of these features and their influence on the model's predictions can serve as a valuable guide for targeted interventions and policy decisions, ultimately leading to improvements in the health, nutrition, and well-being of adolescent girls in Ethiopia. These insights not only validate existing domain knowledge but also evaluate the effectiveness of the model, resulting in more accurate and impactful interventions related to adolescent women health in the region to tackle the intergenerational cycle of the burden of malnutrition. Although the SHAP value is positive, the influence of these factors is slight. This suggests that while residence and contraceptive use are important, they may not be the primary drivers of stunting. Addressing stunting may require a broader approach that combines media campaigns with direct nutritional support, healthcare access, and education on adolescent health practices.

The third objective of the study was to use association rule mining (ARM) with the a priori algorithm. The top seven rules generated by the best model revealed that being 18–19 years old, having a poor wealth index, having no formal education, using an unimproved toilet facility, living in a rural area, having one or more children, not using contraceptive methods, residing in certain regions (Afar, Tigray, Amhara), and having no media exposure were most frequently associated with a high probability of stunting. This finding was supported by previous literatures [25, 57, 83]. Therefore, identifying patterns of features has wide-ranging implications across health and wellbeing, enabling data-driven decision making, improving nutritional status of adolescent, optimizing processes, and supporting research efforts.

The ARM findings indicated that an adolescent girl aged 18–19 are at a higher risk of stunting compared to those aged 15–17, which is in line with research conducted in India [39, 84], this could be attributed to the fact that stunting reflects prolonged exposure to nutrient deficiencies, which may manifest later in life. Additionally, older adolescents are at an increased risk of pregnancy, leading to competition for nutrients with their growing fetus and resulting in various macro and micronutrient deficiencies. Moreover, adolescent girls with poor wealth index were more likely to be stunted compared with their counterparts. This was supported by

studies conducted in Ethiopia [20, 85] and Turkey [86]. This might be because adolescent girls from poor wealth index cannot easily access and afford balanced nutrition, and get nutrition-related information from the media [85].

The likelihood of stunting among adolescent girls in rural areas was found to be higher compared to their urban counterparts, which is consistent with previous research conducted in Ethiopia [20, 85, 87]. This could be attributed to the limited access to healthcare services and lack of exposure to immunization, nutrition information, and educational campaigns in rural areas. Additionally, adolescent girls from the regions of Afar, Tigray, and Amhara were more likely to be stunted compared to girls from other regions, which aligns with a previous study in Ethiopia [40], that identified statistically significant hot spot areas in these regions. This could be linked to seasonal drought and decreased rainfall, which pose challenges for crop production in these regions.

Adolescent girls who have given birth to one or more children were found to have a higher likelihood of being stunted compared to those who have not given birth, which is in line with previous research [20, 88]. This could be attributed to the increased nutritional demands during adolescence, as there is significant competition for nutrients between the still-growing adolescent mother and her rapidly developing fetus. This competition may lead to compromised growth and development for both the mother and the fetus [89]. Besides, the presence of unimproved toilet facilities was found to increase the likelihood of stunting in comparison to adolescents with access to improved latrines, which aligns with previous research [20, 90]. This may be because unimproved toilets can lead to parasitic infections, which are a common cause of malnutrition.

The if/ then rules are critical to discovering hidden relationships between attributes, extracting knowledge from a set of data, and accurately representing knowledge and information about stunting. The current study findings are critically important for policymakers and stakeholders to support public health action, decision-making purposes, and the storage of knowledge regarding adolescent nutritional status. Strategies targeting the identified features should be emphasized.

The practical significance of this study lies in its ability to aid in early detection, provide targeted prevention strategies, and guide personalized interventions, and influence resource allocation and policymaking. These implications have the potential to greatly enhance the health outcomes of adolescent girls in Ethiopia by effectively addressing stunting and reducing its impact on individuals, families, and the healthcare system. As a result, this study introduces new perspectives to the field of stunting intervention among adolescent girls through its innovative approach, identification of key risk factors, development of accurate prediction models, and proposal of personalized interventions. These contributions provide valuable information for policymakers and program planners and offer insightful guidance for designing focused interventions to improve the health outcomes of adolescent girls in Ethiopia and to break the intergenerational cycle of the burden of malnutrition.

## Strength and limitations of the study

The study incorporates eight supervised ML classification algorithms and association rule mining providing a comprehensive and robust analysis of the predictive capabilities of different algorithms in order to reveals hidden patterns and relationships in the data that may not be easily identifiable through traditional statistical methods.

The analysis relied on secondary data from the DHS; crucial clinical, household food security and dietary factors were not taken into account. Consequently, it is crucial for future researchers to incorporate these variables into their datasets when predicting stunting, using

sources other than the DHS. The 2016 dataset may not adequately represent current conditions or trends, limiting the applicability of our findings to today's context. Therefore, future researchers shall use the recent dataset to better predict stunting and provide up-to-date evidences. Besides, the challenges of applying continuous-data methods or machine learning algorithms to discrete variables are also another limitation of our study. Therefore, adapting machine learning algorithms and developing new methods to handle discrete variables effectively and use a complex large data set to utilize generative AI and deep learning are an active area of research in the field. Furthermore, it is important to note that the results of our study may not be generalizable to different populations or age groups, as our investigation specifically targeted adolescent girls in Ethiopia. Therefore, future research should aim to examine stunting classification and prediction across a variety of demographic groups. Besides, biases or limitations could arise from the Boruta feature selection method, only DHS data set used, and limited algorithms included. Therefore, it would be valuable for future research to explore the classification and prediction of stunting using many more algorithms, feature selection methods and multiple data sources to address these limitations and to investigate additional areas that can enhance our understanding of stunting in this population, ultimately guiding more effective interventions and policies.

## Conclusion and implication of the study

This research indicates that the random forest classifier performed better than other algorithms based on performance evaluation metrics. Factors such as region, age, poor wealth index, unimproved toilet facility, rural residence, having children, lack of media exposure, lack of formal education, occupation, religion, and not using contraceptive methods were found to be the top 11 important features in predicting stunting. Therefore, in addition to current efforts to address childhood stunting, nutrition interventions should focus on adolescents as a key target group to break the cycle of malnutrition across generations. More studies using different machine learning algorithms are needed to explore dietary patterns and nutrient intake about adolescent stunting using datasets other than DHS. It is crucial for national strategies to prioritize stunting to reach the most vulnerable individuals in the poorest households, including those with more children and those in the most affected regions. The findings also suggest that interventions to address adolescent stunting in Ethiopia should focus on promoting contraceptive use, increasing media access, reducing poverty, and improving sanitation. Additionally, healthcare workers should screen the nutritional status of adolescent girls and assess risk factors for stunting, with particular emphasis on those living in rural areas, in the poorest wealth quintile, and without access to hygienic toilets.

Our study can have a significant impact on addressing stunting in developing countries. It can enable early detection and diagnosis by analyzing stunting-related data, facilitate remote monitoring and telemedicine to overcome healthcare access limitations, optimize treatment strategies based on patient data, aid in public health planning and resource allocation, recommend personalized interventions, and support data-driven research and policy development [91]. However, successful implementation requires addressing challenges such as data availability, healthcare infrastructure, ethical considerations, and model biases [92]. With proper attention to these challenges, the current study can improve stunting management and outcomes in developing countries.

Policymakers and healthcare providers can use these identified potential factors as indicators to create interventions that meet the specific needs of different subgroups in the population. This tailored approach can enhance the health and nutritional status of adolescent girls and reduce the burden of stunting in areas with limited resources.

## Supporting information

**S1 File. Stunting final dataset.**
(DTA)

## Acknowledgments

This study was based on data from the DHS Program and the authors would like to extend their deepest gratitude to the DHS program data archivist and the Ethiopian CSA.

## Author Contributions

**Conceptualization:** Alemu Birara Zemariam.

**Data curation:** Alemu Birara Zemariam, Mulat Ayele.

**Formal analysis:** Alemu Birara Zemariam, Addis Wondmagegn Alamaw, Habtamu Setegn Ngusie.

**Methodology:** Biruk Beletew Abate, Befkad Derese Tilahun.

**Software:** Alemu Birara Zemariam.

**Writing – original draft:** Alemu Birara Zemariam, Biruk Beletew Abate, Addis Wondmagegn Alamaw, Eyob shitie Lake, Mulat Ayele, Befkad Derese Tilahun, Habtamu Setegn Ngusie.

**Writing – review & editing:** Alemu Birara Zemariam, Gizachew Yilak.

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
