## [Decision Letter · Decision Letter 0]

22 Mar 2024

PONE-D-23-41442Application of Machine Learning-Based Algorithms for Predicting Stunting among Adolescent Girls in EthiopiaPLOS ONE

Dear Dr. Zemariam,

Thank you for submitting your manuscript to PLOS ONE. After careful consideration, we feel that your manuscript will likely be suitable for publication if it is revised to address the points below. Therefore, my decision is "Major Revision".

We look forward to receiving your revised manuscript.

Kind regards,

Oluwafemi Samson Balogun, Ph.D.

Academic Editor

PLOS ONE

Journal Requirements:

https://bmcmedinformdecismak.biomedcentral.com/articles/10.1186/s12911-023-02102-w

https://www.mdpi.com/2227-9067/10/10/1638

https://archpublichealth.biomedcentral.com/articles/10.1186/s13690-015-0093-9

In your revision ensure you cite all your sources (including your own works), and quote or rephrase any duplicated text outside the methods section. Further consideration is dependent on these concerns being addressed.

Reviewers' comments:

Reviewer's Responses to Questions

**Comments to the Author**

1. Is the manuscript technically sound, and do the data support the conclusions?

Reviewer #1: No

Reviewer #2: Yes

2. Has the statistical analysis been performed appropriately and rigorously? 

Reviewer #1: No

Reviewer #2: Yes

3. Have the authors made all data underlying the findings in their manuscript fully available?

Reviewer #1: Yes

Reviewer #2: Yes

4. Is the manuscript presented in an intelligible fashion and written in standard English?

Reviewer #1: No

Reviewer #2: Yes

5. Review Comments to the Author

Reviewer #1: Introduction could potentially be enhanced:

• Expanding on how machine learning has been utilized in public health, especially in developing countries, could provide a more comprehensive backdrop. This includes examples of successful ML applications in similar contexts.

• A detailed discussion about the challenges in predicting stunting, especially in low-resource settings, and how machine learning can address these challenges would be informative. This includes limitations of traditional statistical methods and the advantages offered by ML techniques.

• For readers who may not be familiar with machine learning, a clearer explanation of key ML concepts and algorithms could be beneficial. This includes a basic overview of the algorithms used in the study and why they are suitable for this type of analysis.

• How stunting is influenced by socio-economic and environmental factors, and how ML can help untangle these complex relationships, would provide a more holistic understanding.

• Providing a review of previous studies that have used ML for similar purposes, highlighting what has been achieved and where gaps remain

• Rationale for the Study , particularly the need for ML approaches in the context of Ethiopia.

Other drawbacks

• The absence of some potentially relevant clinical, dietary, and socio-economic variables in the DHS data may affect the accuracy and comprehensiveness of the results.

• The study focuses specifically on adolescent girls in Ethiopia, which may limit the generalizability of the findings to other populations or demographic groups.

• The Boruta algorithm, employed for feature selection in the study, is a robust method that ensures comprehensive inclusion of relevant features. However, it comes with drawbacks such as high computational intensity due to its iterative nature and reliance on random forest classifications, which can be resource-heavy. There's a risk of overfitting, as Boruta tends to retain more features, potentially including noisy or less relevant ones. Its dependence on the performance of the random forest algorithm means its effectiveness is closely tied to how well this model fits the data. Additionally, the process of feature selection can sometimes be arbitrary and lacks clear interpretability, as the reasons for the inclusion or exclusion of certain features aren't always transparent. Furthermore, the efficiency of Boruta can be challenged in handling very high-dimensional data due to the exponential increase in features when creating shadow attributes.

• Machine Learning Model Interpretability can be seen as "black boxes". Include interpretable AI

• ML models parameter optimization is missing

• Include a flowchart to explain the study clearly

• Despite using the Synthetic Minority Oversampling Technique (SMOTE) to balance the data, there's a risk that this technique might introduce artificial bias. Oversampling can sometimes lead to overfitting, as the model might be too tailored to the synthetic examples created by SMOTE. Explain this.

• Missing Data percentage

• The absence of p-values in the summary statistics table of the study could be considered a drawback, particularly in the context of medical and epidemiological research.

• Results interpretation please indicate the clearly the results is based on testing dataset.

• The study uses association rule mining to identify patterns between features and stunting. However, these rules should be interpreted with caution as they indicate correlation, not causation, and the findings might be influenced by confounding factors.

The discussion section of the manuscript, could benefit from a more explicit with a broader and more detailed comparison with existing literature,. Addressing the study's methodological limitations more thoroughly would provide a more balanced view. There's also a need for a more comprehensive discussion on potential biases inherent in the dataset and how these might influence the findings, a more detailed exploration of specific policy recommendations and practical applications would add value. Suggestions for future research, especially addressing the current study's limitations, could provide clearer direction for subsequent studies. Finally, a more in-depth discussion of the socioeconomic, cultural, and environmental factors affecting stunting in the Ethiopian context would offer more understanding of the issue.

Reviewer #2: PONE-D-23-41442

Application of Machine Learning-Based Algorithms for Predicting Stunting among Adolescent Girls in Ethiopia

The manuscript contains an important insight for handling and exploring the non-linear relationship of variables, it is an excellent supplement to results generated using the classical linear statistical models. In general, the manuscript is interesting and relevant to the field.

The following issues needs your reflection and where necessary revisions.

• Do you have a rationale to split the dataset to 80% and 20% for training and testing the model respectively, why not other proportions

• What strategy has employed to mitigate overfitting in your study

• Have you used the same software for data preprocessing and model selection analysis, if not kindly mention which other tools were employed?

• You have mentioned a number of performance evaluation metrics, why don’t you use one of the metrics that best fit to your case.

• Feature selection methods is not well explained, specifically it is important to depict how the employed feature selection method (Boruta algorithms) is suitable over other methods for your study.

• The first paragraph of the discussion part is well explained on the introduction section, no need to repeat it, hence remove it and focus on interpreting your findings and portray its policy and further research implications.

• Random forest algorithm has the best predictive model in your case; your discussion is limited to studies that come across with similar result to yours. But there are studies which identify best predictive model other than random forest algorithm, you have to include them and narrate in relation to your findings.

• Paragraphs narrating the predictors of stunting and the use of association rule mining on the discussion part is direct replication of the result and devoid of interpretation and comparison with other studies’ findings

6. PLOS authors have the option to publish the peer review history of their article (what does this mean?). If published, this will include your full peer review and any attached files.

Reviewer #1: **Yes: **Sorayya Malek

Reviewer #2: No

---

## [Author Response · Author response to Decision Letter 0]

3 Apr 2024

April 1/2024

Point by point response for reviewer’s comments 

Subject: - submission of revised manuscript and point by point response

Manuscript Number: - PONE-D-23-41442

Title: “Application of Machine Learning-Based Algorithms for Predicting Stunting among Adolescent Girls in Ethiopia"

To: - PLOS ONE Journal

Dear editor and reviewers,

First of all, we would like to thank you for giving us the opportunity to made a revision and resubmit a revised manuscript. We appreciate the time and effort that you and the reviewers have dedicated to reviewing and providing constructive, building, and improvable comments that would improve the substance and content of the manuscript. We are very grateful for the insightful comments and valuable inputs to our manuscript. We have considered each comments and clarification questions of reviewer on the manuscript thoroughly. All comments, suggestions, and concerns of the reviewers are clearly stated and well-addressed in our point-by-point responses described in detail on the following pages. Furthermore, the details of changes were shown using the red color on the recently attached revised manuscript. The manuscript language was checked by language professionals and we have followed the journal guidelines upon preparation and submission of the manuscript. Thank you so much again! 

Dear Dr. Zemariam,

Thank you for submitting your manuscript to PLOS ONE. After careful consideration, we feel that your manuscript will likely be suitable for publication if it is revised to address the points below. Therefore, my decision is "Major Revision".

Kind regards,

Oluwafemi Samson Balogun, Ph.D.

Authors’ response: Dear respected academic editor; we thank you for your time, kind words, and special consideration to provide us valuable comments to enhance the quality of our manuscript. We have incorporated all comments, questions and suggestions raised by the reviewers. We kindly request you to see the point by point response and the revised manuscript.

Journal Requirements:

Authors’ response: Dear we thank you and greatly appreciate your valuable suggestion. Rest assured, we have carefully reviewed our manuscript to ensure it complies with the guidelines set by your premier journal.

https://bmcmedinformdecismak.biomedcentral.com/articles/10.1186/s12911-023-02102-w

https://www.mdpi.com/2227-9067/10/10/1638

https://archpublichealth.biomedcentral.com/articles/10.1186/s13690-015-0093-9

In your revision ensure you cite all your sources (including your own works), and quote or rephrase any duplicated text outside the methods section. Further consideration is dependent on these concerns being addressed.

Authors’ response: Dear we sincerely thank you for your diligence in identifying the minor occurrence of overlapping text with the previous publication(s). We greatly appreciate your efforts in bringing this matter to our attention. Ensuring the integrity and originality of our work remains our top priority, and we apologize for any inconvenience caused. We have fully accepted your valuable comment, revised the manuscript accordingly, promptly taken action to rectify it, and provided appropriate citations. We kindly invite you to review the revisions made in the updated manuscript.

Authors’ response: Dear we thank you for your crucial recommendation. We have accepted and revised it accordingly. Kindly see the corrections made in the revised manuscript.

Authors’ response: We sincerely appreciate your valuable suggestions. We have thoroughly reviewed and accepted them, and have been incorporated into the revised manuscript. We kindly request you to review the updated version of the manuscript to see the changes made in response to your vital recommendation.

Reviewers' comments:

Reviewer's Responses to Questions

Comments to the Author

1. Is the manuscript technically sounds, and do the data support the conclusions?

Reviewer #1: No

Reviewer #2: Yes

2. Has the statistical analysis been performed appropriately and rigorously?

Reviewer #1: No

Reviewer #2: Yes

3. Have the authors made all data underlying the findings in their manuscript fully available?

Reviewer #1: Yes

Reviewer #2: Yes

4. Is the manuscript presented in an intelligible fashion and written in standard English?

Reviewer #1: No

Reviewer #2: Yes

5. Review Comments to the Author

Please use the space provided to explain your answers to the questions above. 

Reviewer #1: Introduction could potentially be enhanced:

1. Expanding on how machine learning has been utilized in public health, especially in developing countries, could provide a more comprehensive backdrop. This includes examples of successful ML applications in similar contexts.

Authors’ response: Dear respected reviewer thank you for your imperative suggestion. We agree that providing a more comprehensive backdrop would enhance the quality of our manuscript. In response to your comment, we have accepted and revised the manuscript accordingly. We kindly request you to review the revised manuscript to see the expanded content.

2. A detailed discussion about the challenges in predicting stunting, especially in low-resource settings, and how machine learning can address these challenges would be informative. This includes limitations of traditional statistical methods and the advantages offered by ML techniques.

Authors’ response: Dear reviewer, we thank you for your valuable suggestion. We agree that exploring the limitations of traditional statistical methods and highlighting the advantages offered by machine learning techniques would provide informative insights. We have accepted and expanded the introduction in the revised manuscript to delve into the challenges of stunting prediction, the shortcomings of traditional statistical approaches, and the potential benefits of utilizing machine learning methods in this context. We encourage you to review the updated manuscript to see the revisions made.

3. For readers who may not be familiar with machine learning, a clearer explanation of key ML concepts and algorithms could be beneficial. This includes a basic overview of the algorithms used in the study and why they are suitable for this type of analysis.

Authors’ response: Dear esteemed reviewer, thank you for your vital suggestion. In response to your comment, we have included a more concise explanation of the concepts of machine learning and explained why or based on what criteria we have applied some selected algorithms in the introduction section of the revised manuscript. Moreover, we kindly request you to see the details of the advantage and limitation of each algorithm under the methods section. 

4. How stunting is influenced by socio-economic and environmental factors, and how ML can help untangle these complex relationships, would provide a more holistic understanding.

Authors’ response: Dear reviewer, we thank you for your concrete comment. We have thoroughly addressed your issue in the revised manuscript. Hence, we have expanded to highlight your important issue. 

Firstly, stunting is influenced by socio-economic and environmental factors:

#. Malnutrition: Inadequate access to quality food, especially during critical growth periods, contributes to stunting. Chronic malnutrition is often associated with impoverished socio-economic conditions.

#. Poverty: Families in poverty struggle to provide nutritious food, clean water, and healthcare, leading to higher rates of stunting.

#. Maternal health and nutrition: Malnourished mothers are more likely to give birth to underweight babies at higher risk of stunting. Maternal factors like low height and inadequate prenatal care also contribute.

#. Sanitation and hygiene: Poor sanitation and lack of clean water increase the risk of infections and diseases, impairing nutrient absorption and increasing nutrient requirements.

#. Education and awareness: Lack of knowledge about nutrition, childcare, and hygiene practices contributes to stunting.

#. Healthcare access: Limited access to healthcare facilities, including prenatal care and treatment for illnesses, can contribute to stunting.

#. Environmental factors: Pollution, toxins, and poor air quality can negatively impact child growth and development.

Therefore, addressing stunting requires comprehensive measures, including improving socio-economic conditions, promoting education and awareness, enhancing healthcare services, and improving sanitation and hygiene practices. 

Secondly, machine learning can contribute to unraveling these intricate relationships between socio-economic and environmental factors in stunting by:

A. Feature Selection: Identifies key factors and discards irrelevant variables.

B. Nonlinear Relationships: Captures complex interactions and dependencies.

C. Prediction and Risk Assessment: Predicts stunting likelihood and assesses combined effects of factors.

D. Data Integration: Combines diverse data to uncover hidden patterns.

E. Identifying Complex Interactions: Reveals how factors interact, e.g., income, water access, and education.

F. Targeted Interventions: Identifies at-risk subpopulations for tailored interventions. Machine learning enhances understanding for evidence-based interventions and policies against stunting.

5. Providing a review of previous studies that have used ML for similar purposes, highlighting what has been achieved and where gaps remain

Authors’ response: Dear esteemed reviewer, we thank you for your important comment. We have incorporated a comprehensive review of relevant literature, outlining the key findings, advancements achieved through the application of machine learning and existing gaps and associated limitations. Kindly refer to the revised manuscript.

6. Rationale for the study, particularly the need for ML approaches in the context of Ethiopia.

Authors’ response: Dear esteemed reviewer, we express our gratitude for your concrete comment. We have taken your comments into careful consideration and made the necessary revisions to the manuscript. We appreciate the value you have added through your insightful remarks and kindly invite you to review the revisions incorporated in the updated manuscript. Thank you once again for your valuable input.

Other drawbacks

7. The absence of some potentially relevant clinical, dietary, and socio-economic variables in the DHS data may affect the accuracy and comprehensiveness of the results.

Authors’ response: Dear reviewer, we appreciate your concern and thank you for your nice issue. The authors acknowledged that due to secondary source of date (DHS) some important variables might be missed which have an impact on the model predictive accuracy and comprehensiveness of our results. Therefore, we recommended that future researcher should strive to include a broader range of relevant variables to improve the model prediction using data set other than DHS.

8. The study focuses specifically on adolescent girls in Ethiopia, which may limit the generalizability of the findings to other populations or demographic groups.

Authors’ response: Dear honored reviewer, we sincerely appreciate your concern. While our study provides specific insights into adolescent girls' in Ethiopia, caution should be exercised when applying the results to other contexts. Therefore, we recommend that future researchers should include a broader range of populations other than adolescent girls to improve generalizability and deepen our understanding of the topic.

9. The Boruta algorithm, employed for feature selection in the study, is a robust method that ensures comprehensive inclusion of relevant features. However, it comes with drawbacks such as high computational intensity due to its iterative nature and reliance on random forest classifications, which can be resource-heavy. There's a risk of over fitting, as Boruta tends to retain more features, potentially including noisy or less relevant ones. Its dependence on the performance of the random forest algorithm means its effectiveness is closely tied to how well this model fits the data. Additionally, the process of feature selection can sometimes be arbitrary and lacks clear interpretability, as the reasons for the inclusion or exclusion of certain features aren't always transparent. Furthermore, the efficiency of Boruta can be challenged in handling very high-dimensional data due to the exponential increase in features when creating shadow attributes.

Authors’ response: Dear esteemed reviewer, we sincerely appreciate your critical comment. We completely agree with your point. While Boruta is indeed a powerful tool for feature selection, there can be potential challenges associated with its usage. In our study, we have explored various feature selection methods such as Lasso method, PCA; wrapper methods include forward selection, backward elimination, and recursive feature elimination, correlation-based feature selection, and chi-square test and compared their performance using evaluation metrics to select the best feature selection. Through this analysis, we have found that Boruta is the most effective feature selection method. Furthermore, we acknowledge the challenges posed by Boruta as a limitation of the study. Thank you once again for your valuable comment and kindly see it on the revised manuscript. Here below are some challenges and possible solution taken in our manuscript.

A. High computational cost: The algorithm requires multiple iterations and comparisons with shadow features, which can increase the processing time. Therefore, to mitigate this issue the authors considered reducing the number of iterations or subsampling the dataset. Besides, we utilized parallel processing or distributed computing techniques to speed up the computation.

B. Sensitivity to hyper parameters: Boruta has a few hyper parameters that need to be set, such as the number of random shadow features to create and the threshold for feature importance. To address this challenge, authors had performed hyper parameter tuning using techniques like cross-validation or grid search, or Bayesian optimization, or random search.

C. Handling correlated features: Boruta assumes that features are independent of each other, but in reality, there may be correlations among the features. To address this problem, we had conducted principal component analysis (PCA) and algorithms that explicitly handle feature dependencies, such as recursive feature elimination with cross-validation to identify and reduce the impact of correlated features. 

D. Interpretability of results: Boruta provides a ranking of feature importance, but it does not explicitly provide information on the relationship between features and the target variable. To solve this issue, after applying Boruta authors had conducted additional analysis like data visualization, statistical tests, or domain knowledge to interpret the selected fea

---

## [Decision Letter · Decision Letter 1]

18 Sep 2024

PONE-D-23-41442R1Application of Machine Learning-Based Algorithms for Predicting Stunting among Adolescent Girls in EthiopiaPLOS ONE

Dear Dr. Zemariam,

Thank you for submitting your manuscript to PLOS ONE. After careful consideration, we feel that your manuscript will likely be suitable for publication if it is revised to address the points below. Therefore, my decision is "Minor Revision".

Please submit your revised manuscript within Nov 02 2024 11:59PM. If you will need more time than this to complete your revisions, please reply to this message or contact the journal office at plosone@plos.org. Please include the following items when submitting your revised manuscript:A rebuttal letter that responds to each point raised by the academic editor and reviewer(s). You should upload this letter as a separate file labeled 'Response to Reviewers'.A marked-up copy of your manuscript that highlights changes made to the original version. You should upload this as a separate file labeled 'Revised Manuscript with Track Changes'.An unmarked version of your revised paper without tracked changes. You should upload this as a separate file labeled 'Manuscript'.If applicable, we recommend that you deposit your laboratory protocols in protocols.io to enhance the reproducibility of your results. Protocols.io assigns your protocol its own identifier (DOI) so that it can be cited independently in the future. For instructions see: https://journals.plos.org/plosone/s/submission-guidelines#loc-laboratory-protocols. Additionally, PLOS ONE offers an option for publishing peer-reviewed Lab Protocol articles, which describe protocols hosted on protocols.io. Read more information on sharing protocols at https://plos.org/protocols?utm_medium=editorial-email&utm_source=authorletters&utm_campaign=protocols.

We look forward to receiving your revised manuscript.

Kind regards,

Oluwafemi Samson Balogun, Ph.D.

Academic Editor

PLOS ONE

Journal Requirements:

Reviewers' comments:

Reviewer's Responses to Questions

**Comments to the Author**

1. If the authors have adequately addressed your comments raised in a previous round of review and you feel that this manuscript is now acceptable for publication, you may indicate that here to bypass the “Comments to the Author” section, enter your conflict of interest statement in the “Confidential to Editor” section, and submit your "Accept" recommendation.

Reviewer #3: (No Response)

2. Is the manuscript technically sound, and do the data support the conclusions?

Reviewer #3: Yes

3. Has the statistical analysis been performed appropriately and rigorously? 

Reviewer #3: Yes

4. Have the authors made all data underlying the findings in their manuscript fully available?

Reviewer #3: Yes

5. Is the manuscript presented in an intelligible fashion and written in standard English?

Reviewer #3: No

6. Review Comments to the Author

Reviewer #3: Review report

Review report of research manuscript entitled “Application of Machine Learning-Based Algorithms for Predicting Stunting among Adolescent Girls in Ethiopia”

I would like to say thank you for your invitation to review the above-mentioned manuscript!

General Comments

The paper is technically much sounded and the questions are informative. The statistical analysis of the data is also highly sounded, and the findings are appropriately discussed in the context of previous literature. It meets the requirements of the research community well in the data processing and management steps.

1. Reporting prevalence in the summary section for this particular study is senseless. You are running out of the scope of your title. Try to remove it please.

2. Starting from the title you should identify and incline towards either the model or the application mainly. Based on that the title, background, objectives, and all others will be shaped in that way! As to me, the ultimate objective looks emphasizing the application. My suggested topic/title: “Prediction of stunting and its socioeconomic determinants among adolescent girls in Ethiopia using machine learning algorithms”

3. Why are you interested to study only adolescent girls? Give a strong justification in your introduction. Do you think that stunting is the only concern of adolescent girls?

4. What was the mechanism that can be used to manage the lack of transparency and interpretability of the data?

5. What were the possible bias and limitations for your study? In addition, how could you handle it? Try to incorporate these issues in your study clearly.

6. Why did you exclude other parts of undernutrition? Why do you insisted to stunting only? Please give a strong justification.

7. Some of very relevant factors of stunting cannot be available in the secondary data including DHS data. How did you handle such issues? Try to incorporate in your limitation.

8. Eight machine-learning algorithms were included for model building and comparison for this particular study. What additional models did you plan in your future studies? What challenges did you face to be restricted to these eight models only for this study? What are the major effects for your study regarding variations of machine learning algorithms?

9. EDHS 2016 is old aged data. Why did you prefer it? There are recent EDHS data in other countries. Have you consider this issue?

10. To keep the logical coherence and flow of ideas, I recommend you to mention the consequences and effects of stunting following to the prevalence of stunting.

11. Why did you re categorize wealth index?

12. I appreciate your mean SHAP value report and Waterfall plot analysis. However, I have not seen any interpretation for such wonderful parts of your effort. I strongly recommend you to interpret it in detail using the log odd value for each specific findings to make it clear for readers.

13. Make sure that all of your pertinent findings have been discussed very well including its implications.

14. Artificial intelligence is the current active area of research particularly for health data. You have particularly employed machine learning for this specific study. Why did you prefer machine-learning parts of artificial intelligence on top of other artificial intelligence options like generative artificial intelligence and deep learning?

15. Try to revise editorial and grammatical issues throughout your entire document.

Generally, this research addressed untouched and active research area using multiple machine learning algorithms and large data set. After correcting the given comments, this study is qualified for publication, potentially contribute significant role for the entire scientific community.

7. PLOS authors have the option to publish the peer review history of their article (what does this mean?). If published, this will include your full peer review and any attached files.

Reviewer #3: No

---

## [Author Response · Author response to Decision Letter 1]

30 Sep 2024

To: - PLOS ONE Journal

Dear editor and reviewers,

First of all, we would like to thank you for giving us the opportunity to made a revision and resubmit a revised manuscript. We appreciate the time and effort that you and the reviewers have dedicated to reviewing and providing constructive, building, and improvable comments that would improve the substance and content of the manuscript. We are very grateful for the insightful comments and valuable inputs to our manuscript. We have considered each comments and clarification questions of reviewer on the manuscript thoroughly. All comments, suggestions, and concerns of the reviewers are clearly stated and well-addressed in our point-by-point responses described in detail on the following pages. Furthermore, the details of changes were shown using the red colour on the recently attached track change revised manuscript. The manuscript language was checked by language professionals and we have followed the journal guidelines upon preparation and submission of the revised manuscript. Thank you so much again!

---

## [Decision Letter · Decision Letter 2]

12 Dec 2024

Prediction of Stunting and Its Socioeconomic Determinants among Adolescent Girls in Ethiopia using Machine Learning Algorithms

PONE-D-23-41442R2

Dear Dr. Alemu Birara Zemariam,

We’re pleased to inform you that your manuscript has been judged scientifically suitable for publication and will be formally accepted for publication once it meets all outstanding technical requirements.

Kind regards,

Oluwafemi Samson Balogun, Ph.D.

Academic Editor

PLOS ONE

Additional Editor Comments (optional):

Reviewers' comments:

Reviewer's Responses to Questions

**Comments to the Author**

1. If the authors have adequately addressed your comments raised in a previous round of review and you feel that this manuscript is now acceptable for publication, you may indicate that here to bypass the “Comments to the Author” section, enter your conflict of interest statement in the “Confidential to Editor” section, and submit your "Accept" recommendation.

Reviewer #3: All comments have been addressed

2. Is the manuscript technically sound, and do the data support the conclusions?

Reviewer #3: Yes

3. Has the statistical analysis been performed appropriately and rigorously? 

Reviewer #3: Yes

4. Have the authors made all data underlying the findings in their manuscript fully available?

Reviewer #3: Yes

5. Is the manuscript presented in an intelligible fashion and written in standard English?

Reviewer #3: Yes

6. Review Comments to the Author

Reviewer #3: The authors have addressed all of my comments.

It is now acceptable for publication with the current form.

7. PLOS authors have the option to publish the peer review history of their article (what does this mean?). If published, this will include your full peer review and any attached files.

Reviewer #3: **Yes: **Ali Yimer

---

## [Editor Report · Acceptance letter]

16 Dec 2024

PONE-D-23-41442R2 

PLOS ONE

Dear Dr. Zemariam, 

I'm pleased to inform you that your manuscript has been deemed suitable for publication in PLOS ONE. Congratulations! Your manuscript is now being handed over to our production team.

Kind regards, 

on behalf of

Dr. Oluwafemi Samson Balogun 

Academic Editor

PLOS ONE